



# Separating autotrophic and heterotrophic soil CO₂ effluxes in afforested peatlands

Renée Hermans[15], Rebecca McKenzie[26], Roxane Andersen[2], Yit Arn Teh[3], Neil Cowie[4] and Jens-Arne Subke[1]

[1]Department of Biological and Environmental sciences, University of Stirling, Stirling, UK
[2]Environmental Research Institute, University of Highlands & Islands, Thurso, UK
[3]School of Natural and Environmental Sciences, Newcastle University, Newcastle, UK
[4]Centre for Conservation Science, Royal Society for the Protection of Birds Scotland, Edinburgh, UK
[5]IUCN UK Peatland Programme, Edinburgh, UK
[6]Geography Department, Loughborough University, Loughborough, UK

*Correspondence to*: renee.kerkvliet-hermans@iucn.org.uk

**Abstract.** Peatlands are a significant global carbon (C) store, which can be compromised by drainage and afforestation.
Quantifying the rate of C loss from peat soils under forestry is challenging, as soil CO₂ efflux includes both CO₂ produced
from heterotrophic peat decomposition and CO₂ produced by tree roots and associated fungal networks (autotrophic respiration). We experimentally terminated autotrophic belowground respiration in replicated forest plots by cutting through all living tree roots ("trenching"), and measured soil surface CO₂ flux, litter input, litter decay rate and soil temperature and moisture over two years. Annual peat decomposition (heterotrophic CO₂ flux) was $115 \pm 16$ g C m$^{-2}$ y$^{-1}$, representing c. 40% of total soil respiration. Decomposition of needle litter is accelerated in the presence of an active rhizosphere, indicating a
priming effects by labile C inputs from roots. This suggests that our estimates of peat mineralization in our trenched plots are conservative, and underestimate overall rates of peat C loss. Considering also input of litter from trees, our results indicate that the soils in these 30 year-old drained and afforested peatlands are a net sink for C, since substantially more C enters the soil as organic matter, than is decomposed heterotrophically. However, the C balance for these soils should be taken over the lifespan of the trees, in order to determine if the soils under these drained and afforested peatlands are a sustained sink of C, or become
a net source over longer periods of forestry.

## 1 Introduction

Large peatland areas in the boreal and temperate zone have been drained and afforested in Western Europe, especially in the UK, Ireland and the Fennoscandia region, with conifers replacing native peatland vegetation (Andersen et al., 2016). In the UK alone, >800,000 ha (approximately 20%) were affected by this land-use change (Andersen et al., 2016). As well as causing
habitat loss, drainage and afforestation of peatlands influences peatland hydrology, biogeochemistry, and carbon (C) storage.
In Fennoscandia, drainage to improve tree growth on naturally forested peatlands has been shown to cause significant changes



in decomposition and greenhouse gas (GHG) fluxes (Ojanen et al., 2013). However, the effects of drainage and afforestation on temperate peatlands that were previously dominated by smaller-statured vegetation (e.g. blanket bogs, moorland, heathland) are more uncertain (Sloan et al., 2018). While changes in hydrology and soil redox potential are anticipated to accelerate soil

C loss and alter the composition of GHG emissions, there is a very little quantitative data on how this shift in land-use changes soil C dynamics and GHG emissions in temperate peatlands (Hermans et al., 2018). Since northern hemisphere peatlands are estimated to store a third of global terrestrial carbon (Gorham, 2010), changes linked to drainage and afforestation of temperate peatlands could have significant impacts on regional C dynamics and GHG balances.

There is a broad consensus that peatland drainage accelerates the loss of endogenous peat C stocks, but the impacts
of drainage and afforestation on total soil C stocks and soil respiration are less certain (Hargreaves et al., 2003; Mayer et al., 2020; Simola et al., 2012; Vanguelova et al., 2019; Zerva and M. Mencuccini, 2005). This is because our conceptual and numerical models of peat C dynamics are based on studies where drainage is assumed to be the principal human intervention, and there is no effective functional shift in plant community composition linked to land-use change (e.g. bogs drained to improve the productivity of the existing plant community to increase the quantity of forage for livestock). Afforestation of
previously open peatlands in the British Isles (e.g. Mazzola et al., 2020) and Fennoscandia (e.g. Tolvanen et al., 2020) differ from other types of drained peatlands because this land-use change involves a wholesale shift in the functional composition of the plant community (i.e. replacement of short-statured vegetation with trees), leading to potential interactions or synergistic effects arising from changes to both soil hydrology and plant community composition.

The comparatively high rates of net primary production and larger stature of the trees on drained and afforested peatlands can
represent a significant net ecosystem C sink, and consequently represent a large source of detrital material to soil (Yamulki et al., 2013). Thus, it is possible that this input of C from more highly productive trees could partially or wholly offset some of the losses of C derived from mineralization of the original peat. Moreover, changes to soil organic matter (SOM) chemistry due to a shift towards inputs of more recalcitrant and nutrient-poor coniferous litter could further enhance soil C storage in afforested peatlands in the British Isles and Fennoscandia. This is because the trees planted in these regions (e.g. Sitka spruce)
produce tissues that are of poorer quality than the organic matter (OM) generated by the short-statured vegetation that they have replaced (i.e. larger proportion of woody debris generated with higher C:N ratios and greater proportion of recalcitrant compounds like lignin) (Hermans et al., 2018). This may affect soil C stocks by inhibiting or slowing overall rates of decay due to the tree litter's chemical recalcitrance (Liski et al., 2002), but has not been studied in tree plantations on deep peat so far.

In order to close these knowledge gaps, we need to determine the C balance of drained and afforested peatland soils, tracking C inputs from tissue turnover (e.g. litterfall), as well as C losses from SOM decay. However, separating the direct and indirect effects of trees on peat mineralization is methodologically challenging (Subke et al., 2006). Soil $CO_2$ efflux includes both $CO_2$ released from peat mineralization (heterotrophic $CO_2$ flux) and $CO_2$ produced from root-rhizosphere (i.e. autotrophic) respiration; experimental manipulation of autotrophic C supply to the rhizosphere allows a separation of these
two main component fluxes, but introduces a number of potential artefacts (see Subke et al., 2006 for a methods overview



including method-related assumptions and artefacts). Root trenching, where roots are severed throughout the depth of the rooting zone to create areas of forest floor with no recent C input from trees to roots or associated mycorrhizas, has been used across many forest sites, and provides useful insights into relative contributions of autotrophic and heterotrophic soil CO2 efflux and the respective temporal dynamics. This disruption of root exudation and continuous root turnover could influence

peat decay through processes such as microbial priming effects (Kuzyakov, 2006; Subke et al., 2006), where microbial activity (linked with an active rhizosphere) in the soil is stimulated by the addition of easily accessible C from roots (Kuzyakov, 2006). Further experiments are required to evaluate the effects of these other plant-facilitated processes on peat decay, to better constrain estimates of heterotrophic activity in soil.

In this study, we experimentally determined the C budget of a drained and afforested peat soil. Carbon inputs were

tracked by quantifying litterfall, while C outputs were determined using a trenching technique to partition root-rhizosphere and heterotrophic respiration. In addition, we conducted a root decomposition study in rhizosphere and living root-free (i.e. trenched) soil, to constrain artefacts associated with the trenching and in order to determine if rhizosphere-linked processes (e.g. priming effects) facilitated OM decay. We hypothesised that 1) the soils under these forest plantations are a net C source, 2) soil CO2 efflux is dominated by autotrophic respiration and 3) interactions between C supply to the rhizosphere by trees

result in higher decomposition rates of the litter due to rhizosphere priming effects.

## 2 Methods

### 2.1 Study site

The research took place in RSPB's Forsinard Flows National Nature Reserve in the north of Scotland (58° 22' N, 3° 53' W). Four paired plots were established in the beginning of June 2014 in three separate forestry plantation blocks of identical age

containing a mixture of Sitka Spruce (*Picea sitchensis)* and Lodgepole Pine (*Pinus contorta)*. The plantations were drained around 30 years old and very dense (about 5000 trees per ha), with no vascular plant understory, but sporadic patches of moss, predominantly feather moss, e.g. Hypnum jutlandicum, Hylocomium splendens and in some instances, Sphagnum fallax and S. capillifolium in furrows. The average diameter at breast height (DBH) for Sitka Spruce was 13.3 cm and for Lodgepole Pine 17.9 cm, with an average ratio per area of Sitka Spruce : Lodgepole Pine of 0.6. The average canopy cover was 76.3% (Smith

and Hancock, 2016). Peat depths in these three forestry blocks varied between 30 and 260 cm, with depths at research plots between 137 and 204 cm (Smith et al., 2014). The average annual precipitation in the research area between 1981-2010 was 970.5 mm with an average air temperature of 11.4°C measured at the Kinbrace weather station approximately 20 km from the plots (Location: 58º13'89''N, 3º55'1.2''W; Altitude: 103 m above mean sea level; Met Office, 2018). Seasonal averaged water table relative to ground surface is -350 (±20) mm in spring (March-May), -457 (±34) mm in summer (June-August), -404

(±49) mm in autumn (September-November) and -244 (±14) mm in winter (December-February).





## 2.2 Experimental set up

Candidate locations for trenched and control areas in each plot were initially identified at random and soil surface respiration measured. Based on respiration results, two closely matched plots were selected, and randomly allocated a treatment (trenching or control). Paired plots were no more than 10 metres apart from each other.

The double ploughing of the peat at the time of afforestation created a regular micro topography with low-lying furrows (c. 1.5 m wide) flanked by high ridges (plough throws; c. 0.75 m wide) on either side. In between two plough throws, there is an area of c. 0.50 m width of the original (unploughed) surface (Figure 1). The height from the bottom of the furrow to the top of the plough throw is on average 0.5 m and from the original surface to the plough throw is about 0.15 - 0.2 m. In general, conifer seedlings were planted on the plough throws because of the improved drainage compared to the original surface. For our study,

each plot encompassed the three types of micro-topography.

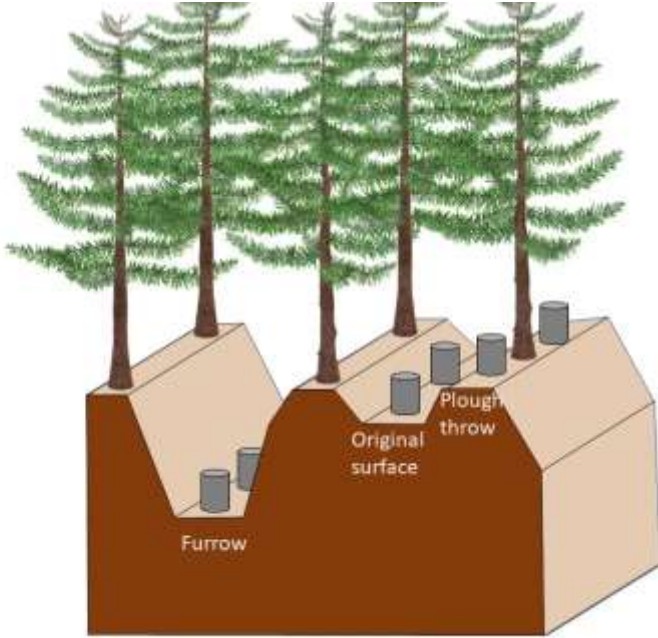

Figure 1 Micro topography in forest plantations, with location of measurements for flux chambers (not to scale).

**2.3 Trenching**

A 40 cm deep and 30 cm wide trench was cut to just below the main rooting depth of the trees, cutting through all roots present. The trench was double-lined using polypropylene gardening cloth, and re-filled with peat soil in between the two layers of cloth to prevent in-growth of roots (Figure 2). The dimensions of each trench plot were about 3.5 x 1.5 meters and included





all three micro topographic forms. These dimensions maximised the space between trees, with closest trees located about 30
cm from trenches.

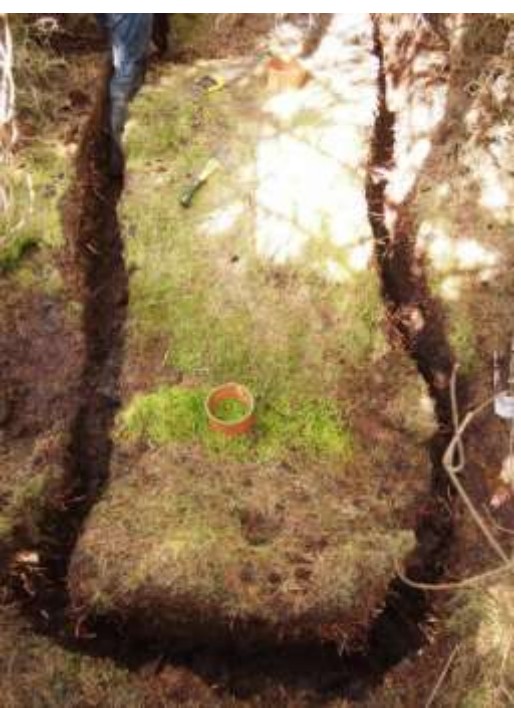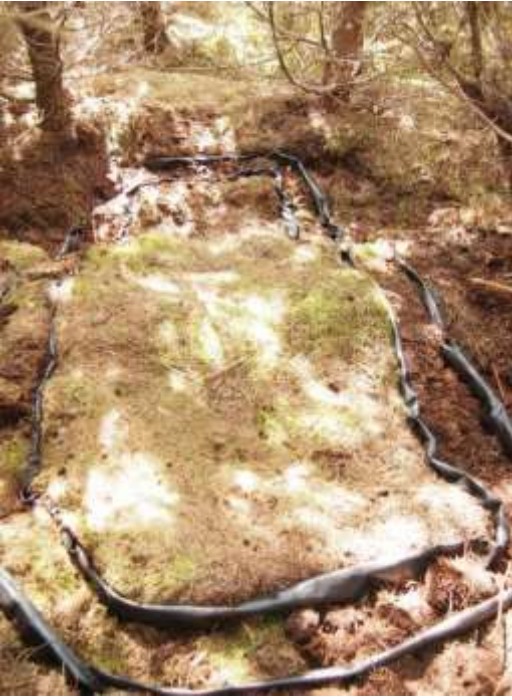

Figure 2 Left: trenching plot dug and all living tree roots are sawn trough. Right: finished trench plot; trenches double lined with polypropylene gardening cloth and filled in with peat.

**2.4 CO₂ measurements**

Three pairs of PVC collars of 10 cm height with a diameter of 20 cm where installed to a depth of three cm within the three microforms (Figure 1) of both trenched and control plots. $CO_2$ measurements were taken using custom-built dark dynamic closed chamber with a height of five cm and a diameter of 20 cm, which were placed on the permanent collars for three minutes. Elasticated rubber material placed over the joint of chambers and collars for the duration of the measurement provided
an air-tight seal. The chamber was connected to an EGM 4 Infrared Gas Analyser (PP-Systems, Amesbury, MA, USA), recording $CO_2$ concentrations every 4-5 seconds. Fluxes were calculated from increases in $CO_2$ concentration within the chamber over 3 minutes. Measurements were carried out ten times between August 2014 and July 2016.

**2.5 Litter**

Six litter traps (0.07 m² each) were located close to each plot, and litter (falling needles and twigs) collected at each sampling
visit. Dry weight of all litter was recorded as area-based averages for each plot separately.



Litter was allowed to fall onto the soil surface within collars for the duration of the experiment. To be able to distinguish peat respiration from litter respiration, surface litter was removed manually from one (always the same) of the two paired collars in each microform before measuring respiration. The litter present in the "collar with litter" was weighed after a respiration measurement and then placed back in the collar. Litter adjacent to the collar was also collected and weighed in the field, then

taken back to the lab, dried and weighed again to establish the wet to dry mass ratio of litter and calculate litter dry mass within each collar.

### 2.6 Roots

Root biomass was determined from soil cores (0-20 cm deep and 6.5 cm diameter) taken from each microform in all plots, at the start (June 2014) and end (July 2016) of the experiment. Roots from each core were carefully separated and sorted into

three root diameter classes: smaller than 2 mm, 2 to 5 mm, and greater than 5 mm. All roots and the root-free soil were dried at 50°C for 7 days, and weighed to establish percentage roots per gram of soil. Root depth was found to be 25 cm when digging trenches.

        To estimate root decomposition directly, roots were taken from soil collected in each plot, dried (50°C for 7 days) and separated in the same size classes as described previously. Between 0.36 and 0.69 g of dried root material (separately for

each size class) were placed in polyester mesh bags (10 x 10 cm; mesh size of 0.5 mm) for field incubations. Bags were soaked in water for 2 days prior to field placement, to mimic field conditions. Four replicate bags of each size class where buried at 5-10 cm depth in all three microforms in all plots four weeks after trenching. To account for any weight loss that may have occurred prior to field incubation, five bags of each size class where taken into the field and not buried, but taken back to the lab; the proportional mass loss of litter in these bags was used to correct the initial root mass of all other bags.

One bag per root class per microform was collected from all sites in November 2014, March 2015, July 2015 (except root class >5 mm, since there was not enough material for four bags) and July 2016. After retrieval, bags were dried for seven days at 50°C, and root dry mass recorded.

Root decay was fitted to an exponential decay function:

$$M_t = M_0 e^{-kt} \tag{1}$$

With $M_t$ the remaining amount of root biomass after collection from the field, $M_0$ the initial root biomass, t time and k the decay constant. Data fits were performed separately for root size and microform.

### 2.7 Soil moisture and temperature

Between June 2014 and July 2016, soil moisture and soil temperature at 5 and 20 cm soil depth were recorded at 30-minute intervals in all three microforms in a nearby plot, using 12-bit smart temperature sensors, S-TMB-M002 (Onset Computer

Corporation, Bourne, MA, USA) and 10HS soil moisture smart sensors, S-SMD-M005 (Decagon Devices, Inc., Pullman, WA,



USA combined with Onset's smart sensor technology) connected to a Hobo micro station (Onset Computer Corporation, Bourne, MA, USA).

In addition to this, soil temperature (10 cm thermistor) and moisture (HH2 moisture meter, Delta-T Devices, Cambridge) were measured at 5 cm depth next to each collar during sampling. Air temperature was also measured one meter above the ground during sampling.

## 2.8 Statistical analyses and flux calculations

Data were analysed using R (RStudio Team, 2016). All $CO_2$ data was log transformed to meet the criteria of normality, and a linear mixed-effect model (LMM) using the *nlme* package in R (Pinheiro et al., 2017) was used to predict $CO_2$ fluxes based on environmental drivers. All numerical predictors were standardized to one standard deviation prior to statistical analyses, to allow relative effect sizes of predictors to be compared directly (Nakagawa and Schielzeth, 2010). Model selection was done based on information theory (Burnham and Anderson, 2002). First a model of maximum complexity was built, with interactions between soil moisture, soil temperature, trench treatment and microform plus interactions between trench treatment, microform and litter treatment, with plot as a random effect. Then, all possible combinations of this model were identified using the 'dredge' function in the *MuMIn* package (Barton, 2017). Goodness of model fit was assessed with the small-sample size corrected Akaike's Information Criterion (AICc), which is calculated using the number of parameters and either the maximum likelihood estimate for the model or the residual sum of squares. "Likelihood" here is a measure of the extent to which a sample provides support for particular values of a parameter in a parametric model. AICc values of different models can be compared and the model with the lowest AICc is selected as the 'best approximating model', hereafter called 'top model' (Burnham and Anderson, 2002). Any of the models with a delta AICc of less than 2 are considered to be as good as the best model (Richards, 2005). 'Dredge' also gives a weight to the models it produces, ranging between 0 and 1; with for example a weight of 0.7 meaning that there is a 70% chance that that model is the best approximating model of the models considered. If the weight of the best model is low, it is not possible to say that that model really is the best model, meaning other models also have to be considered. In this study, the model with the best AICc and highest weight was used. Significance (p-values) for the mixed effect model were calculated using the package *lmerTest* (Kuznetsova et al., 2016).

Annual fluxes were calculated using the predict function over the mixed effects model from library *lme4* in R (Bates et al., 2015). The predictions were made over half-hourly measurements of soil moisture and soil temperature at 5 cm soil depth in all three microforms just outside the plots.

From these predictions, partitioned fluxes were calculated from the collars without litter as:

$$F_a + \varepsilon = (F_{control} + \varepsilon) - ((F_{trench} + \varepsilon) - (F_{dead\ roots} + \varepsilon)) \tag{2}$$

Where $F_a$ is autotrophic $CO_2$ flux, $F_{control}$ is the $CO_2$ flux from the control plots, $F_{trench}$ is the $CO_2$ flux from trenched plots, $F_{dead\ roots}$ is the $CO_2$ flux coming from the dead roots in the trenched plots created by the trenching technique and $\varepsilon$ the associated error terms.





The annual flux from litter was calculated from the difference in modelled annual fluxes between collars with and
without litter.

## 3 Results

### 3.1 Temporal trends in soil $CO_2$ fluxes

Trenching initially led to an increase in soil respiration, followed by a significant reduction in soil $CO_2$ flux. Soil respiration
fluxes from both control and trenched plots showed a clear annual cycle, with highest fluxes in summer. After the initial
perturbation fluxes from trenched plots are significantly lower than fluxes from control plots ($p<0.001$) and this difference is
greater in the summer (Figure 3).

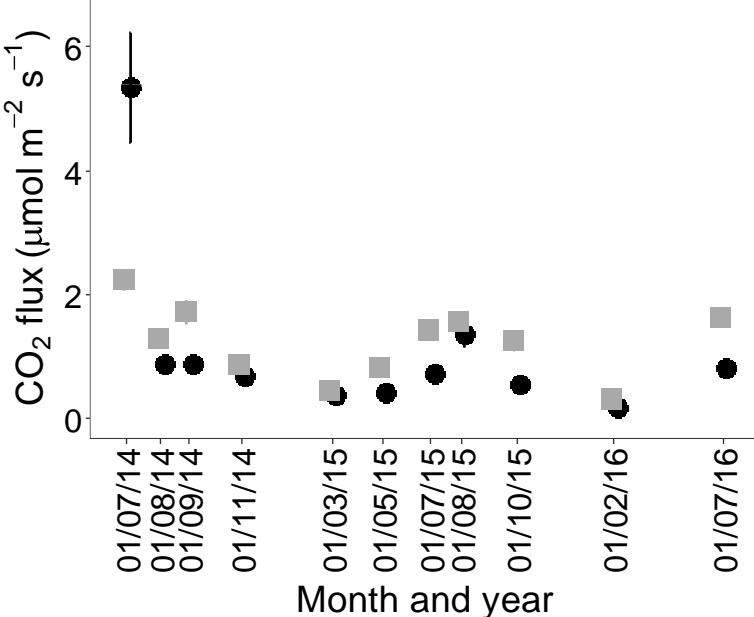

Figure 3 Mean $CO_2$ fluxes from control (grey squares) and trenched (black circles) plots over time, averaging across all
microforms (n = 12) and both litter treatments. Error bars are standard errors and are in most instances smaller than symbols.

Soil $CO_2$ fluxes were best explained with a combination of soil moisture, soil temperature, trenching treatment,
microform and litter treatment, with an interaction between soil moisture and soil temperature, including 'plot' as a random
effect. Table 1 shows model estimates for each variable, with their standard error and p-value. The marginal $R^2$ was 0.40 and
conditional $R^2$ was 0.41. The set of models with a $\Delta$AICc of less than 2 is shown in





Table 2. The predictors of the 'top model' were present in at least over half of the models in the top model set, so this model was used.

Table 1 Model estimates with standard errors and p-value for best-fit model.

| Fixed effect | Estimate | Std. Error | p-value |
|---|---|---|---|
| Intercept | -0.22 | 0.10 | 0.05 |
| Trenched | -0.50 | 0.07 | <0.001 |
| Microform - Original surface | 0.42 | 0.12 | <0.001 |
| Microform - Plough throw | 0.35 | 0.13 | 0.006 |
| Soil moisture | -0.12 | 0.06 | 0.03 |
| Soil temperature | 0.35 | 0.03 | <0.001 |
| No Litter | -0.17 | 0.06 | 0.008 |
| Soil moisture x Soil temperature | -0.11 | 0.04 | 0.008 |

Table 2 Model selection summary, showing the 4 best ranked models with a delta AICc of less than 2. Models are ranked by AICc and weight, where higher weighted models have more statistical support. Df= degrees of freedom, Loglik=Log likelihood, and ΔAICc is the differece in AICc to the 'top model'.

| Candidate models | Df | LogLik | AICc | ΔAICc | Weight |
|---|---|---|---|---|---|
| Trenched + Microform+ No Litter + Soil moisture + Soil temperature + Soil moisture x Soil temperature | 10 | -403 | 826 | 0.00 | 0.34 |
| Trenched + Microform+ No Litter + Soil moisture + Soil temperature | 9 | -404 | 826 | 0.37 | 0.29 |
| Trenched + Microform+ Soil moisture + Soil temperature + Soil moisture x Soil temperature | 9 | -404 | 827 | 1.21 | 0.19 |





| | | | | |
|---|---|---|---|---|
| Trenched + Microform+ Soil moisture + Soil temperature | 8 | -405 | 827 | 1.30 | 0.18 |


### 3.2 Spatial trends and litter contributions to soil $CO_2$ flux

In both control and trenched plots, fluxes from plough throw (control: $1.23 \pm 0.10$; trenched: $0.85 \pm 0.07$ µmol m$^{-2}$ s$^{-1}$ ; p=0.01) and original surface (control: $1.48 \pm 0.10$; trenched: $0.83 \pm 0.06$ µmol m$^{-2}$ s$^{-1}$ ; p<0.001) were significantly higher than fluxes from furrow (control: $0.90 \pm 0.06$; trenched: $0.47 \pm 0.04$ µmol m$^{-2}$ s$^{-1}$). Across all microforms fluxes from collars with litter

($1.03 \pm 0.05$ µmol m$^{-2}$ s$^{-1}$) were significantly higher than fluxes from collars without litter ($0.91 \pm 0.05$ µmol m$^{-2}$ s$^{-1}$, p=0.008).

### 3.3 Role of environmental drivers in modulating $CO_2$ flux

Observed soil $CO_2$ efflux values correlated positively with higher soil temperatures, whilst soil moisture showed an inconsistent correlation with flux values; a significant (p=0.008) interaction between soil temperature and soil moisture means that at high temperatures $CO_2$ flux decreases with increasing soil moisture, but at low temperatures flux increases when soil

moisture increases (Figure 4).

There was no difference in soil temperature between trenched and control plots, but soil moisture was significantly higher in trenched plots than in control plots (p<0.001). To account for the artificially elevated soil moisture in trenched plots, $CO_2$ fluxes were corrected using the global relationship between soil moisture and $CO_2$ flux (Figure 4) to calculate fluxes expected to occur under moisture conditions in control plots.

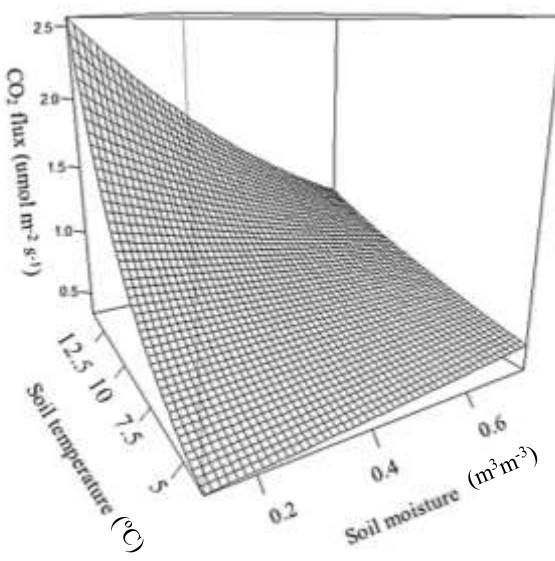






Figure 4 Combined effect of soil temperature and soil moisture at 5 cm depth on soil $CO_2$ flux from the control sites, estimated from the 'top model'.

## 3.4 Partitioned fluxes

Soil $CO_2$ efflux was partitioned into component fluxes for all measuring dates from August 2014 onwards to remove
disturbance related artefacts observed immediately after trenching. Flux simulations based on the soil model details indicate significantly lower autotrophic fluxes than heterotrophic fluxes (p=0.01, Figure 5). Across all microforms, heterotrophic fluxes represented 61% and autotrophic fluxes represented 39% of the total fluxes. From these predictions, annual sums for autotrophic and heterotrophic fluxes have been calculated, giving an average peat decomposition flux of $184 \pm 21$ g C m$^{-2}$ y$^{-1}$. Total soil respiration without needle litter is $301.3 \pm 34.2$ g C m$^{-2}$ y$^{-1}$, with needle decomposition adding $41.2 \pm 53.5$ g C m$^{-2}$
y$^{-1}$ to annual fluxes, giving a total soil respiration including needle litter of $343 \pm 35$ g C m$^{-2}$ y$^{-1}$.









Figure 5 Modelled and measured fluxes of heterotrophic (black) and autotrophic (grey) soil $CO_2$ efflux from the three topographic microforms. Closed symbols are average fluxes with error bars (n = 4 plots). Connecting lines are the predicted fluxes using soil temperature and moisture at 5 cm depth. a) Original surface, b) Plough throw and c) Furrow.

### 3.5 Impact of litter and roots

Litter fall was $719 \pm 71.3$ grams of litter per $m^{-2} y^{-1}$, with no detectable difference between trenched and control plots. Assuming a C fraction of biomass of 50% (Mathews, 1993), this represents an input to the soil of 359 g C $m^{-2} y^{-1}$ via litter fall.

$CO_2$ flux from surface litter is calculated from the difference in the modelled annual fluxes between the collars with and without litter, with the fluxes from collars with litter significantly higher than the $CO_2$ flux from the collars without litter (p = 0.008, Table 1). C emitted by litter in the control plots appears to be higher than in trenched plots. Further, the average amount of litter in the collars (per $m^{-2}$) of the trenched plots is higher than in the collars of the control plots, resulting in a 1.7 to 3.6 times higher $CO_2$ flux per gram of litter from the control plots than from the trenched plots (Table 3).

Table 3 Mean amount of C emitted as $CO_2$ by surface needle litter in 'litter collars', for both years (standard error in brackets).

| Microform | Litter | | $CO_2$ flux | | $CO_2$ flux per g litter | |
|---|---|---|---|---|---|---|
| | [g $m^{-2}$] | | [g C $m^{-2} y^{-1}$] | | [mg C (g litter)$^{-1}$] | |
| | **Trench** | **Control** | **Trench** | **Control** | **Trench** | **Control** |
| **Original surface** | 472.4 (174.0) | 237.8 (61.0) | 34.4 (32.4) | 62.5 (50.4) | 72.8 (73.6) | 263 (222) |
| **Plough throw** | 351.1 (125.2) | 263.5 (85.4) | 36.0 (40.8) | 60.1 (61.3) | 103 (122) | 228 (244) |
| **Furrow** | 589.8 (288.8) | 558.4 (256.2) | 26.4 (28.8) | 43.2 (40.8) | 44.8 (53.5) | 77.4 (81.2) |

Root biomass per $m^2$ with an assumed rooting depth of 25 cm in August 2014 was $761 \pm 324$ g, $603 \pm 110$ g and $715 \pm 257$ g for < 2 mm roots, 2-5 mm roots and > 5 mm roots, respectively. For both the control and the trench plots, roots smaller than 2 mm declined in total biomass from the start of the experiment to the end of the experiment, but there was no significant difference between the control and trenched plots at the start and end of the experiment. Despite an apparent trend towards higher root biomass in control plots in July 2016, these differences were not statistically significant. There was also no significant differences between the beginning and end of the experiment for root classes 2-5 mm and >5 mm (Figure 6).









Figure 6 Root biomass per square meter to 25 cm in control (grey) and trenched (black) plots at the beginning of the experiment (June 2014) and end of experiment (July 2016), split into three root size classes, per microform. a) root size <2 mm, b) root size 2-5 mm, c) root size >5 mm. Symbols are average fluxes with error bars representing standard errors (n = 4 plots).

**3.6 Root decomposition**

Root mass in decomposition bags showed a consistent decline over the duration of the experiment (Figure 7). The calculated decay constant (k) showed systematic differences by microform, with highest decay rates tending to occur for incubations at "original surface" (Table 4).

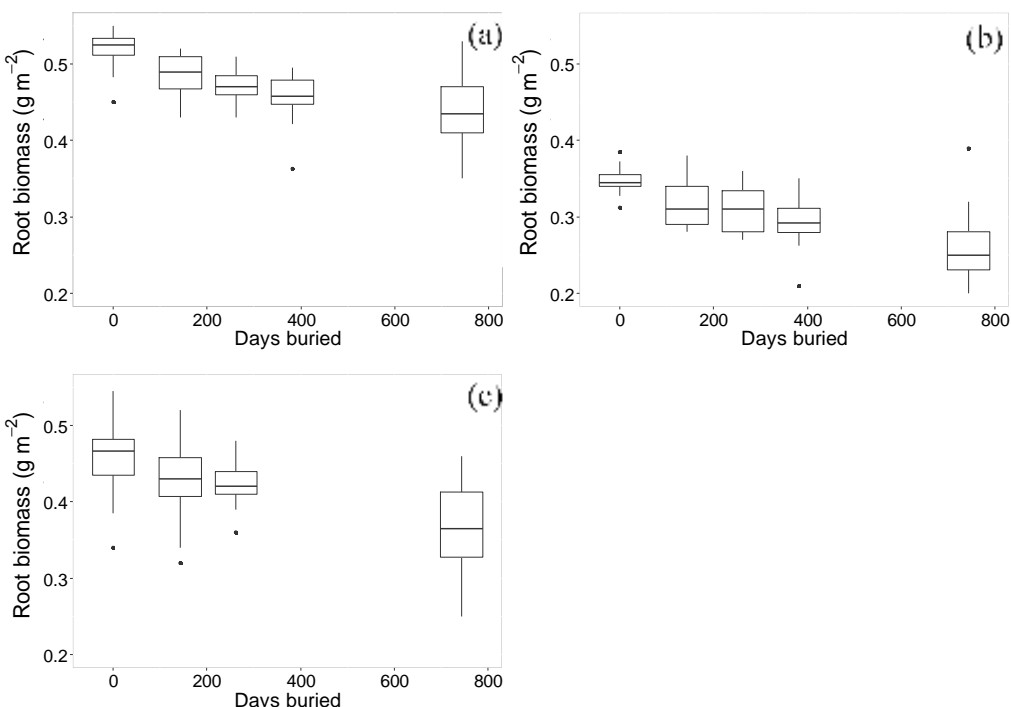

Figure 7 Root mass remaining in root bags over number of days buried, for each root size class averaged across all microforms.
a) root size <2 mm, b) root size 2-5 mm, c) root size >5 mm.





Table 4 Root decay constants and associated C emissions in trenched plots by microform and root size classes (standard error in brackets).

|  | Root diameter mm | Decay constant y⁻¹ | C emitted year 1 g C m⁻² y⁻¹ | C emitted year 2 g C m⁻² y⁻¹ |
|---|---|---|---|---|
| Original surface | <2 | 0.106 (0.019) | 57 (18) | 52 (17) |
|  | 2-5 | 0.180 (0.027) | 28 (7) | 23 (6) |
|  | >5 | 0.190 (0.027) | 50 (25) | 41 (21) |
|  | **Total** | **269 (63)** | **135 (32)** | **116 (27)** |
| Plough throw | <2 | 0.097 (0.015) | 39 (12) | 35 (11) |
|  | 2-5 | 0.153 (0.026) | 33 (6) | 28 (5) |
|  | >5 | 0.100 (0.022) | 10 (1) | 9 (1) |
|  | **Total** | **164 (27)** | **82 (13)** | **73 (12)** |
| Furrow | < 2 | 0.068 (0.015) | 10 (3) | 9 (3) |
|  | 2-5 | 0.097 (0.018) | 11 (4) | 10 (3) |
|  | >5 | 0.065 (0.023) | 3 (/) | 3 (/) |
|  | **Total** | **48 (10)** | **24 (5)** | **22 (5)** |

**3.7 C flux from dead roots**

The amount of C emitted from the decaying roots is calculated using the exponential decay function (1), with the biomass of roots per m² to a depth of 20 cm in the trenched plots at the beginning of the experiment as $M_0$. It is assumed that all biomass lost is emitted as $CO_2$ and that 50% of roots is C (Mathews, 1993), as conservative assumptions, meaning that estimates are maximum possible $CO_2$ flux from decaying roots in trenched plots. To correct for soil $CO_2$ efflux generated in trenched plots as an artefact of creating dead root biomass, annual estimated $CO_2$ emissions were corrected by subtracting root-decay based
estimates from trenched plot $CO_2$ emissions (Table 5).





Table 5 Corrected heterotrophic (peat only; $F_h$) and autotrophic ($F_a$) fluxes (standard error in brackets) in g C m$^{-2}$ y$^{-1}$ for dead root decay in trenched plots for both first (August 2014 – August 2015) and second year (August 2015 – August 2016) of the study.

| | Year 1 | | Year 2 | | Average | | |
|---|---|---|---|---|---|---|---|
| | $F_h$ | $F_a$ | $F_h$ | $F_a$ | $F_h$ | $F_a$ | $F_{soil}$ |
| Original surface | 86.5 (37.7) | 276 (35) | 80.2 (33.2) | 242 (30) | 83.3 (35.4) | 259 (33) | 342 (48) |
| Plough throw | 118 (29) | 212 (20) | 124.1 (28.0) | 199 (19) | 121 (28) | 206 (19) | 327 (34) |
| Furrow | 134 (21) | 125 (10) | 107 (17) | 105 (9) | 120 (19) | 115 (9) | 235 (21) |

Heterotrophic fluxes represents approximately 24% and autotrophic fluxes 76% of the total soil fluxes in the original surface, 37% and 63% respectively in the plough throw, 51% and 49% respectively, in the furrow and 38% and 62% respectively averaged over all microforms.

## 3.8 Weighted average for soil CO$_2$ flux in Flow Country forest plantations

In order to scale soil CO$_2$ fluxes (excluding litter) from different microforms to the level of an entire forest stand, fluxes from individual microforms were weighted by their fractional area (Table 6). This results in a slight shift in proportion of heterotrophic and autotrophic CO$_2$ flux sources to 40% and 60% respectively and a total area weighed soil CO$_2$ flux of 289.4 ± 18.6 g C m$^{-2}$ y$^{-1}$.

Table 6 Area-weighted heterotrophic (peat only) ($F_h$) and autotrophic ($F_a$) fluxes (standard error in brackets) in g C m$^{-2}$ y$^{-1}$ averaged over both years measured.

| Microform | Fractional area | Area-weighted $F_h$ | Area-weighted $F_a$ |
|---|---|---|---|
| Original surface | 0.14 | 12 (5) | 36 (5) |
| Plough throw | 0.43 | 52 (13) | 88 (8) |
| Furrow | 0.43 | 52 (8) | 49 (4) |
| Total | 1 | 115 (16) | 174 (10) |





## 4 Discussion

Mass balance calculations indicate that the soils in these 30 year old drained and afforested peatlands are a net sink for C, as
substantially more C enters the soil as organic matter, than is decomposed heterotrophically. The C balance of the soil under
these forest plantations is visualised in

Figure 8, with the annual $CO_2$ fluxes of the forest plantation based on the area-weighted fluxes.

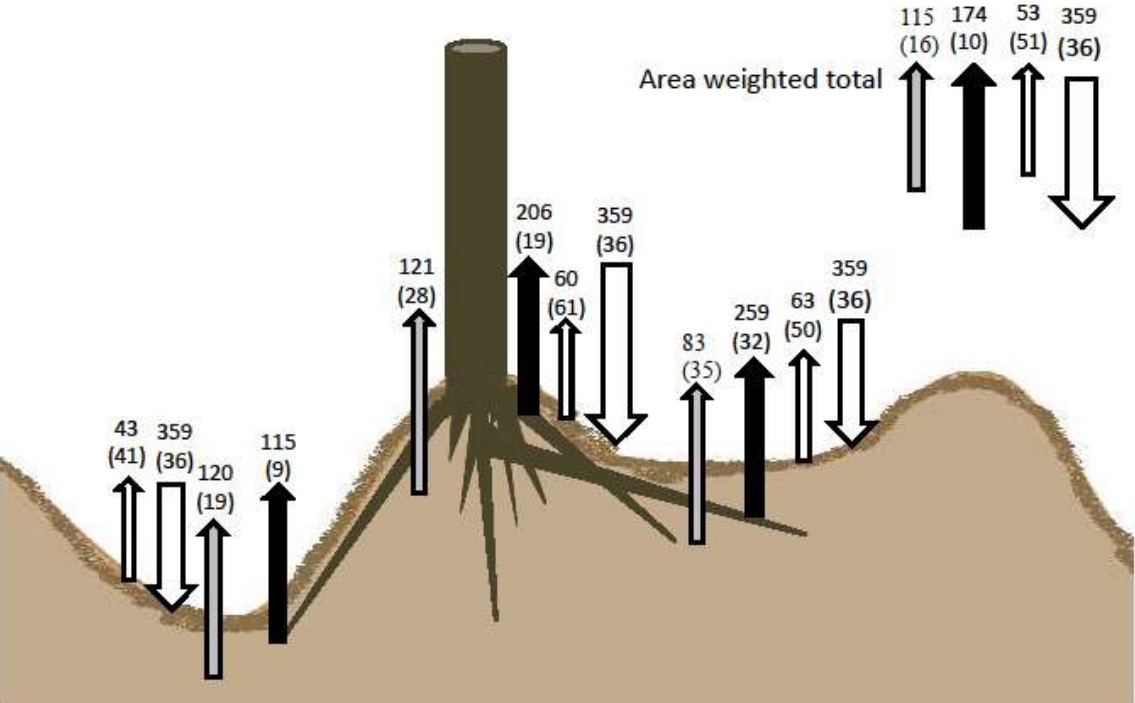

Figure 8 Summary of soil C budget (in g C m$^{-2}$ y$^{-1}$) under the canopy of a Sitka spruce and Lodgepole pine plantation on deep
peat. Downward arrows indicate litterfall, whilst upward fluxes show autotrophic ($F_a$) (black), heterotrophic (peat only; $F_h$)
(grey), and surface needle litter (white) across the three main microforms. Total forest soil fluxes with area weighted values
are visualised in the top right corner.

We found a C input of 359 g C m$^{-2}$ y$^{-1}$ via litter fall, similar to other Sitka Spruce forests of similar age to these forest plantations
in the UK, which range from 273 to 573 g C m$^{-2}$ y$^{-1}$ (Morison et al. 2012). This is balanced by total soil efflux including litter-
derived $CO_2$ of 342.5 ± 34.7 g C m$^{-2}$ y$^{-1}$, i.e. the amount of C entering the soil as surface litter alone falls within a similar range
to C leaving as $CO_2$. Missing in the C budget in

Figure 8 are the input of C from root exudates that remain in the soil C pool (net rhizodeposition), potential losses through
aquatic C export, and root growth and turnover. Gaffney, et al. (2020) have measured the aquatic C flux (DOC, DIC, POC)
from similar forest plantations in the north of Scotland and have found a flux of 13.9 g C m$^{-2}$ y$^{-1}$, which is a small flux compared





to the $CO_2$ soil efflux. Root:shoot allocation in forest ecosystems is usually in the order of 1:3 (Laiho and Laine, 1997), so it is possible that belowground productivity could account for a significant C input only partially sampled by our approach (i.e. excluding large root stocks), adding to these plantations being a potential carbon sink.

The rate of peat decomposition in these drained and afforested peatlands is substantial, but overall soil $CO_2$ efflux and ratio of heterotrophic/autotrophic respiration falls within boreal averages for forested ecosystems (Figure 9; Bond-Lamberty and Thomson, 2010). Average soil efflux corrected for microform area without litter (to determine the peat decomposition rate) over the two measurement years was $289 \pm 19$ g C $m^{-2}$ $y^{-1}$ of which $174 \pm 10$ g C $m^{-2}$ $y^{-1}$ is autotrophic and $115 \pm 16$ g C $m^{-2}$ $y^{-1}$ is heterotrophic.

Our total soil respiration (including litter) of $342.5 \pm 34.7$ g C $m^{-2}$ $y^{-1}$ is slightly higher than values of around 260 g C $m^{-2}$ $y^{-1}$ reported for a similar forest plantation in Ireland, a 39-year old drained Sitka spruce plantation on naturally treeless blanket bog (Byrne and Farrell, 2005). Our peat oxidation rates of $115 \pm 16$ g C $m^{-2}$ $y^{-1}$ are also higher than found by Hargreaves et al. (2003), who found <100 g C $m^{-2}$ $y^{-1}$ in a 26 year old Sitka spruce plantation on drained deep (> 2 m) peat in Scotland. However, they point out that their estimate has a big uncertainty, since it was calculated from the difference between total net C exchange and net tree gain, which both have a large uncertainty.

Comparing our results to a study in 18 to 44 year old Sitka spruce plantations on poorly drained Dystric Histosols in Southern Ireland, our results of total soil respiration is much lower than their 972 g C $m^2$ $yr^{-1}$. This could be due the difference in age of the trees. Our results do show a similar percentage of peat-only respiration of 38% of the total soil respiration minus the litter flux, compared to their 35% (Jovani-Sancho et al., 2018).

Minkkinen et al (2018) measured carbon fluxes of a drained, naturally forested peatland (Kalevansuo peatland forest)
in the south of Finland over 4 years. They found that the ecosystem was a strong sink of $CO_2$ in all studied years, with an average NEE for the 4 years of -234 g C $m^{-2}$ $y^{-1}$. By subtracting the carbon sink of the tree stand from NEE the authors show that the soil was a carbon sink of -60 g C $m^{-2}$ $y^{-1}$. By modelling their forest soil respiration fluxes from chamber measurements they found the peat only respiration made up 53% of the total forest floor respiration flux, litter 22%, roots 16% and autotrophic respiration of above-ground vegetation 8%. Our results show a lower percentage of peat only respiration of 38% of the total
soil respiration minus the litter flux. The higher percentage found by Minkkinen et al (2018) could possibly be explained by them not applying a dead root correction to their fluxes, which as we show could lead to a big difference in fluxes.



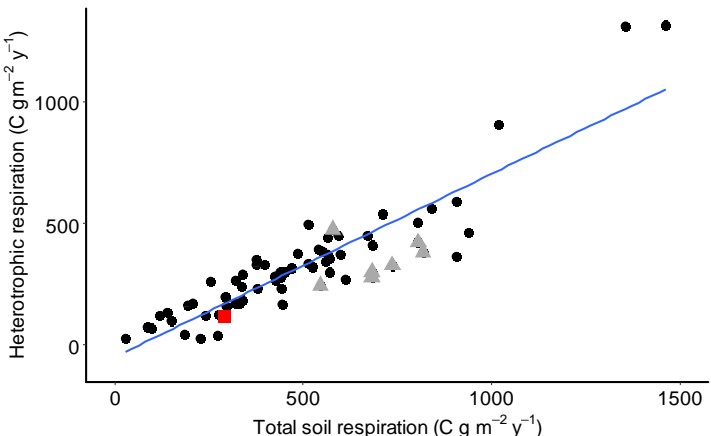

Figure 9 Heterotrophic annual flux against soil respiration annual flux (g C m$^{-2}$ y$^{-1}$) in Boreal forests in black dots, Boreal
forests on peat soils in grey triangles, from Bond-Lamberty and Thomson (2010) dynamic database downloaded on
19/04/2020. Our study is included in the red square. See Table 7 in Appendix for references to studies used to create this graph.

Trenching is likely to underestimate heterotrophic respiration and rates of peat mineralization since this approach
does not account for rhizosphere effects such as priming (Walker et al., 2016). Therefore, it is likely that the estimated rate of
peat oxidation from trenching is conservative. The observed difference of 1.7 to 3.6 times higher $CO_2$ flux per gram of litter
from the control plots than from and trenched plots (Table 3) indicates that heterotrophic processes are reduced under trenching.
In presence of an active rhizosphere (control plots), decomposition of needle litter and/or soil organic matter (SOM) appears
to be faster than when the rhizosphere is not active (trenched plots). This priming of organic matter decomposition is likely to
be the result of microbial activity in the soil stimulated by the addition of easily accessible C from roots (Kuzyakov, 2006).
Further, several studies have shown that mycorrhizal fungi are involved in soil priming (Kuyper, 2017; Paterson et al., 2016).
Therefore, in the control plots a slightly larger proportion of the total $CO_2$ flux could be heterotrophic decomposition than the
fluxes from the trenched plots suggest, which means there could be an underestimation of heterotrophic flux in our results, in
line with results from literature (Subke et al., 2004, 2011).

To calculate the root biomass at the start of the experiment, one soil core per microform was taken and since trees
were standing close to each other (1.5 to 2 m apart) this was assumed to be representative for the whole microform. It was not
possible to distinguish between living and dead roots in the soil cores, so living root biomass might have been overestimated.
The dead root emission correction made a big difference to the ratio of heterotrophic to autotrophic flux, going from 61% and
39% respectively over all microforms to 38% and 62% respectively (without area-weighting of fluxes), so a decrease in
heterotrophic flux of 23%. This is in line with the corrections used in other studies; a comparison of corrections used in
trenching studies indicates a range from 2% to 24%, with an average of 12% (Subke et al., 2006). This big difference in the
fraction heterotrophic : autotrophic flux suggests that even two years after trenching, decaying root biomass make significant
contributions to the $CO_2$ flux.

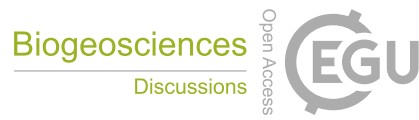

That soils under a 30-year old conifer plantation on deep peat are a C sink is an interesting finding. However, in order to determine if these soils are a long-term sink or source of carbon, the balance between soil C input from roots and litter and C loss via oxidation should be quantified over the lifespan of the plantation. The average peat depth in these forest plots is 126 (±16) cm, with 0.47 kg C m$^{-2}$ per centimetre depth (Cannell et al., 1993), which represents ca. 59.3 (±7.3) kg C m$^{-2}$ stored in the peat under these plantations. The largest carbon losses most likely occurred in the initial planting phase of afforestation (Simola et al., 2012; Vanguelova et al., 2019; Zerva and M. Mencuccini, 2005), but fluxes have not been measured through this phase and cannot be quantified as new policy prevents planting on deep peat (Forestry Commission, 2016). In other parts of the world, peatland drainage is still actively happening and studies from these sites show very high rates of peat oxidation during the first 5-10 years of conversion (e.g. McCalmont et al., 2021).

## 5 Code and data availability

The R code and datasets analysed in this paper are not publicly available. Requests to access the datasets should be directed to renee.kerkvliet-hermans@iucn.org.uk.

## 6 Author contribution

Funding acquisition and initial conceptualization of the whole PhD project was done by J-A Subke, R. Andersen, Y. Teh and N. Cowie. Further in-depth conceptualization of this particular work was done by R. Hermans and J-A Subke, with support from R. Andersen, Y Teh and N. Cowie. Investigation, methodology, project administration, data curation, formal analysis, software, visualization and writing was done by R. Hermans with supervision and guidance of J-A Subke, R. Andersen, Y. Teh and N. Cowie.

## 7 Competing interests

The authors declare that they have no conflict of interest.

## 8 Acknowledgements

We are grateful for the PhD studentship that enabled this work, jointly funded by the University of Stirling and the Royal Society for the Protection of Birds. Trenching of plots was supported by volunteers working at the RSPB Forsinard reserve and the Environmental Research Institute of the University of the Highlands and Islands. We thank Rebecca McKenzie, Paul Gaffney, Peter Gilbert and Nathalie Triches for help with field measurements. Further we gratefully acknowledge Daniela Klein, Norrie Russell, Trevor Smith (RSPB), Graeme Findlay (Forestry Commission Scotland), Fountains Forestry and Donald MacLennan (Brook Forestry) for guidance and land access.



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



## 10 Appendix

**Table 7 List of articles used in Figure 9**

| Authors | Year | Title |
|---|---|---|
| Bond-Lamberty, B, C Wang, and ST Gower | 2004 | Contribution of Root Respiration to Soil Surface CO2 Flux in a Boreal Black Spruce Chronosequence |
| Bond-Lamberty, Ben, Dustin Bronson, Emma Bladyka, and Stith T. Gower | 2011 | A Comparison of Trenched Plot Techniques for Partitioning Soil Respiration |
| Gaumont-Guay, D., T. A. Black, A. G. Barr, T. J. Griffis, R. S. Jassal, P. Krishnan, N. Grant, and Z. Nesic. | 2014 | Eight Years of Forest-Floor CO2 Exchange in a Boreal Black Spruce Forest: Spatial Integration and Long-Term Temporal Trends |
| Gaumont-Guay, David, T. Andrew Black, Tim J. Griffis, Alan G. Barr, Rachhpal S. Jassal, and Zoran Nesic | 2006 | Interpreting the Dependence of Soil Respiration on Soil Temperature and Water Content in a Boreal Aspen Stand |
| Gaumont-Guay, David, T. Andrew Black, Tim J. Griffis, Alan G. Barr, Kai Morgenstern, Rachhpal S. Jassal, and Zoran Nesic | 2006 | Influence of Temperature and Drought on Seasonal and Interannual Variations of Soil, Bole and Ecosystem Respiration in a Boreal Aspen Stand |
| Gaumont-Guay, David, T Andrew Black, Alan G Barr, Rachhpal S Jassal, and Zoran Nesic | 2008 | Biophysical Controls on Rhizospheric and Heterotrophic Components of Soil Respiration in a Boreal Black Spruce Stand |
| Hermle, Sandra, Michael B. Lavigne, Pierre Y. Bernier, Onil Bergeron, and David Paré | 2010 | Component Respiration, Ecosystem Respiration and Net Primary Production of a Mature Black Spruce Forest in Northern Quebec |
| Högberg, Peter, Anders Nordgren, Nina Buchmann, Andrew F. S. Taylor, Alf Ekblad, Mona N. Högberg, Gert Nyberg, Mikaell Ottosson-Löfvenius, and David J. Read | 2001 | Large-Scale Forest Girdling Shows That Current Photosynthesis Drives Soil Respiration |



| Howard, Erica A., Stith T. Gower, Jonathan A. Foley, and Christopher J. Kucharik | 2004 | Effects of Logging on Carbon Dynamics of a Jack Pine Forest in Saskatchewan, Canada |
| Laganière, Jérôme, David Paré, Yves Bergeron, and Han Y.H. Chen | 2012 | The Effect of Boreal Forest Composition on Soil Respiration Is Mediated through Variations in Soil Temperature and C Quality |
| Ma, Y., Piao, S., Sun, Z., Lin, X., Wang, T., Yue, C., and Yang, Y. | 2014 | Stand ages regulate the response of soil respiration to temperature in a Larix principis-rupprechtii plantation |
| Mäkiranta, Päivi, Kari Minkkinen, Jyrki Hytönen, and Jukka Laine | 2008 | Factors Causing Temporal and Spatial Variation in Heterotrophic and Rhizospheric Components of Soil Respiration in Afforested Organic Soil Croplands in Finland |
| Mahli, Y, D D Baldocchi, and P G Jarvis | 1999 | The Carbon Balance of Tropical, Temperate and Boreal Forests |
| Molchanov, A. G. | 2009 | $CO_2$ Emission from the Surface of Dark Gray Forest Soils of the Forest Steppe and Sandy Soddy-Podzolic Soils of the Southern Taiga |
| Mustamo, P., Maljanen, M., Hyvärinen, M., Ronkanen, A. K., and Kløve, B. | 2016 | Respiration and emissions of methane and nitrous oxide from a boreal peatland complex comprising different land-use types. |
| O'Conell, K. E. B., Gower, S. T., & Norman, J. M. | 2003 | Net Ecosystem Production of Two Contrasting Boreal Black Spruce Forest Communities |
| Ojanen, Paavo, Kari Minkkinen, Jukka Alm, and Timo Penttil | 2010 | Soil-Atmosphere $CO_2$, $CH_4$ and $N_2O$ Fluxes in Boreal Forestry-Drained Peatlands |
| Ojanen, Paavo, Kari Minkkinen, Annalea Lohila, Tiina Badorek, and Timo Penttilä | 2012 | Chamber Measured Soil Respiration: A Useful Tool for Estimating the Carbon Balance of Peatland Forest Soils? |
| Pumpanen, J., Kulmala, L., Lindén, A., Kolari, P., Nikinmaa, E., and Hari, P. | 2015 | Seasonal dynamics of autotrophic respiration in boreal forest soil estimated by continuous chamber measurements |
| Russell, C A, and R P Voroney | 1998 | Carbon Dioxide Efflux from the Floor of a Boreal Aspen Forest. I. Relationship to Environmental Variables and Estimates of C Respired |



| | | |
|---|---|---|
| Ryan, M. G., M. B. Lavigne, and S. T. Gower | 1997 | Annual Carbon Cost of Autotrophic Respiration in Boreal Forest Ecosystems in Relation to Species and Climate |
| Sawamoto, Takuji, Ryusuke Hatano, Masato Shibuya, Kunihide Takahashi, Alexander P. Isaev, Roman V. Desyatkin, and Trofim C. Maximov | 2003 | Changes in Net Ecosystem Production Associated with Forest Fire in Taiga Ecosystems, near Yakutsk, Russia |
| Vogel, JG;, DW; Valentine, and RW Ruess | 2005 | Soil and Root Respiration in Mature Alaskan Black Spruce Forests That Vary in Soil Organic Matter Decomposition Rates |
| Widén, Britta, and Hooshang Majdi | 2001 | Soil $CO_2$ Efflux and Root Respiration at Three Sites in a Mixed Pine and Spruce Forest: Seasonal and Diurnal Variation |
| Zha, Tianshan, Zisheng Xing, Kai Yun Wang, Seppo Kellomaki, and Alan G. Barr | 2007 | Total and Component Carbon Fluxes of a Scots Pine Ecosystem from Chamber Measurements and Eddy Covariance |