# Peer review of "Separating autotrophic and heterotrophic soil CO2 effluxes and net soil carbon balance in afforested peatlands"

_Biogeosciences, 2021_

## Referee Comment (RC1)

**Referee comments on MS No. bg-2021-126: 'Separating autotrophic and heterotrophic soil CO2 effluxes in afforested peatlands'**

**General comments**

This study helps answer an important, highly policy-relevant question concerning use of peatlands in temperate regions for plantation forestry. Very limited research on the implications for climate change of this land use on this soil type has been published. This work provides empirical data to support modelling of the balance between CO2 emission due to peat decomposition and atmospheric CO2 removal into tree biomass. It clarifies the reliability of assumptions used about the relative rates of heterotrophic and autotrophic restoration to estimate the rate of peat decomposition from total soil CO2 efflux and will inform similar assumptions in future. It highlights the important role of rhizosphere priming effects in decomposition of afforested peat. This study is excellent - well conceived, carefully undertaken and concisely reported. Its limitations are recognised and discussed.

**Specific comments**

1. Your finding that the soil of these 30-year-old forests is a net C sink is arguably as important as the findings about the relative magnitudes of the autotrophic and heterotrophic CO2 effluxes. The title of the preprint indicates a focus on the latter. Consider expanding discussion of the net soil C balance and altering the title to reflect a dual focus.

2. The likelihood that killing roots by trenching will also have stopped rhizosphere priming of peat decomposition is acknowledged as a limitation of the study. The priming of litter decomposition in the same way is demonstrated to make a substantial difference to litter-derived CO2 efflux by the litter decomposition measurements in the trenched and control plots but no evidence is provided on the likely size of this effect on peat decomposition. Any further evidence that can be obtained from the literature would help in assessing the degree of underestimation of peat decomposition by the trenching treatment.

3. Generally, you have been consistent about the boundaries of the system under study (line 74: 'the C budget of a drained and afforested peat soil'). Mentions of root growth in line 324 and belowground productivity in line 327 are slightly confusing because assimilation of C in tree biomass was not included in your study. If by 'root growth and turnover' and 'belowground productivity' you are referring to root litter and/or exudate deposition, make this clearer. It is important that readers do not confuse soil C stocks with below-ground C stocks.

4. The limitations of not measuring fluvial C fluxes or root litter and exudate deposition are briefly mentioned but could be discussed more fully in the context of their implications for afforested peatland soil C balance. These limitations and any conclusion about their likely implications for the main findings should be mentioned briefly in the abstract.

5. The final discussion point about the importance of knowing the net C balance over the lifespan of a plantation is important and welcome but, for balance, should be expanded. The fact that this lifespan normally ends with timber harvesting and deposition of large quantities of felling residues above ground and whole root systems below ground means that we need to go beyond a single forestry rotation to assess the soil C balance of the land use. The separate litter and dead root decomposition fluxes reported here may help inform assessment of post-felling $CO_2$ fluxes but need to recognise the different water table level and soil moisture conditions created by the soil rewetting associated with clear-felling.

**Technical corrections**

18. Consider adding a sentence saying that you measured and corrected for decomposition of the excised roots.

31. 'treed' perhaps better than forested as these can be quite sparse.

35. 'very little' better than 'a very little'.

51. Consider adding 'or outstrip' after 'could partly or wholly offset'.

52. 'original' could be omitted.

55. 'of poorer quality' could be replaced by 'less readily decomposed'.

58. Consider omitting 'chemical', the recalcitrance is also biological and due partly to the microbial environment of the peat.

61. Consider adding 'rhizodeposition' after 'litterfall'. Although you didn't measure it, it's important not to hide the fact that it occurs and needs to be considered as a C input to soil.

78. 'forestry plantations' is probably a better description of the land use/ecosystem than 'forest plantations'.

79-80. Something missing in Hypothesis 3. 'Interactions between C supply to the rhizosphere by trees' and what?

85. Insert a comma after 'drained'.

89. Is there a simpler way of saying 'with an average ratio per area of Sitka spruce : Lodgepole pine of 0.6'? Perhaps omit this and insert '3:5 (on average)' before 'mixture' in line 85.

92. 11.4 $^{\circ}$C is the 30-year average maximum temperature. It would be better to give the average mean temperature or if not available, also give the average minimum, which is 3.3 $^{\circ}$C for Kinbrace.

97-99. Excellent approach.

100. 'double ploughing' is ambiguous. Either say 'double-mouldboard ploughing', which is technically correct or 'twin-throw, spaced-furrow ploughing' which is perhaps more universally iunderstood.

100-105. In section 2.2, could you reduce the text description to a single sentence by including the dimensions in Figure 1?

114-115. Consider replacing 'with closest trees located about 30 cm from trenches' by 'but did not represent ground within 30 cm of trees'. But if 30 cm was the distance from trees to the outer edge of the trench, the unrepresented ground would be that within 60 cm of trees.

133-134. Say how you distinguished the litter that had fallen since the previous measurement.

138-139. Say that the roots extracted from the soil cores and weighed included both live and dead roots.

142. Say if you assumed that root density in the 20-25 cm soil layer was the same as in the 0-20 cm layer.

144. Replace 'in' with 'into'.

213. remove 'at least' or 'over'.

227. Change 'higher soil temperatures' to 'soil temperature'.

229-230. If possible, state the soil temperature above which $CO_2$ efflux decreases with soil moisture.

241-243. Having already read the abstract, this was slightly confusing. It is clearly explained in the discussion (372-373) but can you add a few words here to emphasise that the model prediction of heterotrophic respiration includes that for decomposition of excised roots?

257-258. The increased litterfall into collars in the trenched treatment compared with the control is potentially interesting but is not supported by the litter trap catches. Say if the difference is significant but omit from the text if not.

263-264. Could be worth mentioning possible bias from not sampling ground close to trees.

267. Replace 'was' with 'were'.

283. Table 4. It is unclear what the figures in bold in the 'Decay constant' column are. If they are 2-year excised root-derived C emissions, consider moving them to a new column added on the right of the table or omit them altogether.

303. Replace 'weighed' with 'weighted'.

308. Table 6. Area-weighted fluxes and the breakdown into autotrophic and heterotrophic fluxes are quite sensitive to the area fractions of the different microforms. I checked these against some measurements I had for double-mouldboard ploughing at another northern Scotland site and found them to be quite different. Please double check that these are correct.

310. Omit comma after 'matter'.

313. Nice diagram. Would it be possible to add a net C balance figure for each microform and the area-weighted total?

316. Omit comma after '(grey)'.

324. As mentioned in Specific comment 3 above, consider replacing 'root growth and turnover' with ' root litter deposition' or 'rhizodeposition' (the latter would include root exudates).

327. As mentioned in Specific comment 3 above, consider replacing 'belowground productivity' with 'belowground litter and exudate deposition'.

341. Replace 'Southern Ireland' with 'the south of Ireland or just 'Ireland'.

361. Replace 'and' with 'the'.

370-371. The first half of the sentence is an important 'Methods' detail so I've suggested adding it around lines 138-139. If you do that, you might need to reword this sentence slightly.

376-377. This is an interesting finding that could inform the assessment of soil $CO_2$ fluxes after timber harvesting on afforested peatland. Consider adding it to the abstract.

384-386. A single citation of a tropical peatland study is not very helpful here. Consider either omitting this last sentence or citing a wider range of evidence, preferably as relevant as possible to temperate afforested peatlands.

---

## Author Comment (AC1)

**Referee 1**

This study helps answer an important, highly policy-relevant question concerning use of peatlands in temperate regions for plantation forestry. Very limited research on the implications for climate change of this land use on this soil type has been published. This work provides empirical data to support modelling of the balance between CO2 emission due to peat decomposition and atmospheric CO2 removal into tree biomass. It clarifies the reliability of assumptions used about the relative rates of heterotrophic and autotrophic restoration to estimate the rate of peat decomposition from total soil CO2 efflux and will inform similar assumptions in future. It highlights the important role of rhizosphere priming effects in decomposition of afforested peat. This study is excellent - well conceived, carefully undertaken and concisely reported. Its limitations are recognised and discussed.

Thank you for this positive appraisal of our study.

**Specific comments**

1. Your finding that the soil of these 30-year-old forests is a net C sink is arguably as important as the findings about the relative magnitudes of the autotrophic and heterotrophic CO2 effluxes. The title of the preprint indicates a focus on the latter. Consider expanding discussion of the net soil C balance and altering the title to reflect a dual focus.

We agree that the net C sink in afforested peatlands is an important finding, and have expanded and in part re-written the discussion around this (also in response to another referee; see below). We also agree that the net C balance of soils should be reflected in the title, and have changed it to: "Separating autotrophic and heterotrophic soil CO2 effluxes and net soil carbon balance in afforested peatlands"

2. The likelihood that killing roots by trenching will also have stopped rhizosphere priming of peat decomposition is acknowledged as a limitation of the study. The priming of litter decomposition in the same way is demonstrated to make a substantial difference to litter-derived CO2 efflux by the litter decomposition measurements in the trenched and control plots but no evidence is provided on the likely size of this effect on peat decomposition. Any further evidence that can be obtained from the literature would help in assessing the degree of underestimation of peat decomposition by the trenching treatment.

This is a good point, and we have extended the discussion on this point, adding new insights from a recent study where tree planting on heathlands was demonstrate to result in loss of organic surface soil (Lines 384-387).

3. Generally, you have been consistent about the boundaries of the system under study (line 74: 'the C budget of a drained and afforested peat soil'). Mentions of root growth in line 324 and belowground productivity in line 327 are slightly confusing because assimilation of C in tree biomass was not included in your study. If by 'root growth and turnover' and 'belowground productivity' you are referring to root litter and/or exudate deposition, make this clearer. It is important that readers do not confuse soil C stocks with below-ground C stocks.

We removed the term "root growth" to now only mention "root turnover" in line 328 to avoid confusion between living plant C stocks and belowground litter production. However, we think that "belowground productivity" is meaningful in this context, as greater productivity results in more turnover and hence organic matter (i.e. litter) input to the soil.

4.  The limitations of not measuring are briefly mentioned but could be discussed more fully in the context of their implications for afforested peatland soil C balance. These limitations and any conclusion about their likely implications for the main findings should be mentioned briefly in the abstract.

We have extended the discussion around flux estimates and absence of direct belowground litter input, and also added the following sentence to the abstract (lines 24-26): "This study doesn't account for fluvial C fluxes, which represents a small flux compared to the $CO_2$ soil efflux; further, root litter and exudate deposition could be a significant C source that is only partially sampled by our approach, adding to these plantations being a potential carbon sink."

5.  The final discussion point about the importance of knowing the net C balance over the lifespan of a plantation is important and welcome but, for balance, should be expanded. The fact that this lifespan normally ends with timber harvesting and deposition of large quantities of felling residues above ground and whole root systems below ground means that we need to go beyond a single forestry rotation to assess the soil C balance of the land use. The separate litter and dead root decomposition fluxes reported here may help inform assessment of post-felling CO2 fluxes but need to recognise the different water table level and soil moisture conditions created by the soil rewetting associated with clear-felling.

"In the UK forest plantations on deep peat usually end in clear felling of the site and restoration of the peat. The results of this study could also help inform what the $CO_2$ fluxes will be when timber is harvested and large quantities of felling residues are left above ground as well as whole root systems below ground. However, we note that changes in water table and soil moisture conditions created by the soil rewetting associated with clear-felling will have significant and separate impacts beyond the conditions of active drainage under which we took our measurements."

**Technical corrections**

18. Consider adding a sentence saying that you measured and corrected for decomposition of the excised roots.

Added: "Decomposition of cut roots was measured and $CO_2$ fluxes were corrected for this."

31. 'treed' perhaps better than forested as these can be quite sparse.

Changed to "treed".

35. 'very little' better than 'a very little'.

We deleted "a".

51. Consider adding 'or outstrip' after 'could partly or wholly offset'.

Added "or outstrip".

52. 'original' could be omitted.

We deleted "original"

55. 'of poorer quality' could be replaced by 'less readily decomposed'.

Replaced with "are less readily decomposed".

58. Consider omitting 'chemical', the recalcitrance is also biological and due partly to the microbial environment of the peat.

61. Consider adding 'rhizodeposition' after 'litterfall'. Although you didn't measure it, it's important not to hide the fact that it occurs and needs to be considered as a C input to soil.

Added: "and rhizodeposition"

78. 'forestry plantations' is probably a better description of the land use/ecosystem than 'forest plantations'.

Changed to "forestry plantations"

79-80. Something missing in Hypothesis 3. 'Interactions between C supply to the rhizosphere by trees' and what?

We have added "and surface litter decomposition"

85. Insert a comma after 'drained'.

Done

89. Is there a simpler way of saying 'with an average ratio per area of Sitka spruce : Lodgepole pine of 0.6'? Perhaps omit this and insert '3:5 (on average)' before 'mixture' in line 85.

We added in 3:5 (on average) and deleted 'with an average ratio per area of Sitka spruce : Lodgepole pine of 0.6'

92. 11.4 $^\circ$C is the 30-year average maximum temperature. It would be better to give the average mean temperature or if not available, also give the average minimum, which is 3.3 $^\circ$C for Kinbrace.

Changed to "an average maximum air temperature of 11.4°C and average minimum air temperature of 3.3°C"

97-99. Excellent approach.

Thank you!

100. 'double ploughing' is ambiguous. Either say 'double-mouldboard ploughing', which is technically correct or 'twin-throw, spaced-furrow ploughing' which is perhaps more universally understood.

Changed to "double-mouldboard ploughing"

100-105. In section 2.2, could you reduce the text description to a single sentence by including the dimensions in Figure 1?

We considered adding dimensions to Figure 1, but think that this won't help clarity, and would prefer to retain the text as it is.

114-115. Consider replacing 'with closest trees located about 30 cm from trenches' by 'but did not represent ground within 30 cm of trees'. But if 30 cm was the distance from trees to the outer edge of the trench, the unrepresented ground would be that within 60 cm of trees.

Changed to "but did not represent ground within 60 cm of trees".

133-134. Say how you distinguished the litter that had fallen since the previous measurement.

We didn't make this distinction, as litter turnover that includes fresh litter was part of our experimental approach. By weighing actual litter amounts in collars, we were able to calculate the $CO_2$ flux coming from the litter.

138-139. Say that the roots extracted from the soil cores and weighed included both live and dead roots.

Added "Both dead and living roots".

142. Say if you assumed that root density in the 20-25 cm soil layer was the same as in the 0-20 cm layer.

Added in "and root density for 0-20 cm was assumed to be representative for 0-25 cm".

144. Replace 'in' with 'into'.

Done

213. remove 'at least' or 'over'.

Deleted "at least".

227. Change 'higher soil temperatures' to 'soil temperature'.

Done

229-230. If possible, state the soil temperature above which CO2 efflux decreases with soil moisture.

We now indicate in the text that the relationship has an inflection temperature between 6 and 7 °C

241-243. Having already read the abstract, this was slightly confusing. It is clearly explained in the discussion (372-373) but can you add a few words here to emphasise that the model prediction of heterotrophic respiration includes that for decomposition of excised roots?

Added: "The model prediction of heterotrophic respiration includes that for decomposition of cut roots."

257-258. The increased litterfall into collars in the trenched treatment compared with the control is potentially interesting but is not supported by the litter trap catches. Say if the difference is significant but omit from the text if not.

263-264. Could be worth mentioning possible bias from not sampling ground close to trees.

We're unsure why there should be a bias from proximity of trees, as root sampling included locations close to trees, representing root densities of the stand appropriately.

267. Replace 'was' with 'were'.

Done

283. Table 4. It is unclear what the figures in bold in the 'Decay constant' column are. If they are 2-year excised root-derived C emissions, consider moving them to a new column added on the right of the table or omit them altogether.

We apologise for this. The bold numbers in the first column were left from a previous version of the table and we forgot to take them out. This is corrected, bold numbers in the other column are the totals (and labelled as such).

303. Replace 'weighed' with 'weighted'.

Done

308. Table 6. Area-weighted fluxes and the breakdown into autotrophic and heterotrophic fluxes are quite sensitive to the area fractions of the different microforms. I checked these against some measurements I had for double-mouldboard ploughing at another northern Scotland site and found them to be quite different. Please double check that these are correct.

Here the furrows where c. 1.5 m wide, plough throws c. 0.75 m wide on either side of the original surface, which was c. 0.5 m wide. This gives the area fractions used in table 6.

310. Omit comma after 'matter'.

Done

313. Nice diagram. Would it be possible to add a net C balance figure for each microform and the area-weighted total?

Thank you. We have added the net C balance for the area weighted flux only, since we think it would become too difficult to read diagram otherwise.

316. Omit comma after '(grey)'.

Done

324. As mentioned in Specific comment 3 above, consider replacing 'root growth and turnover' with

' root litter deposition' or 'rhizodeposition' (the latter would include root exudates).

As indicated above, we have adjusted the text as suggested to make a clearer distinction between root stocks and turnover.

327. As mentioned in Specific comment 3 above, consider replacing 'belowground productivity' with 'belowground litter and exudate deposition'.

We also made this amendment as suggested.

341. Replace 'Southern Ireland' with 'the south of Ireland or just 'Ireland'.

Replaced with "the south of Ireland".

361. Replace 'and' with 'the'.

Done

370-371. The first half of the sentence is an important 'Methods' detail so I've suggested adding it around lines 138-139. If you do that, you might need to reword this sentence slightly.

Changed to: "so our results might overestimate the living root biomass"

376-377. This is an interesting finding that could inform the assessment of soil CO2 fluxes after timber harvesting on afforested peatland. Consider adding it to the abstract.

Added in abstract: "Decomposition of cut roots was measured and $CO_2$ fluxes were corrected for this, this resulted in a big change in the fraction heterotrophic : autotrophic flux, suggesting that even two years after trenching decaying root biomass make significant contributions to the $CO_2$ flux."

384-386. A single citation of a tropical peatland study is not very helpful here. Consider either omitting this last sentence or citing a wider range of evidence, preferably as relevant as possible to temperate afforested peatlands.

We have added further recent papers focussing on peatland drainage.

---

## Author Comment (AC2)

This is a neat study, carefully designed and carried out, with important implications for modelling afforested peatland systems. For the most part, the work is clearly described and well written. Some areas which need clarification are listed below. My main comment is that the significance of the finding that the peat is a net sink for carbon 30 years after being drained and afforested is very much under-played. This is contrary to expectations and current modelling assumptions, so merits more discussion.

> Thank you for this positive feedback on our study. We agree that the net C sink in afforested peatlands is an important finding, and have expanded and in part re-written the discussion around this (also in response to another referee; see below).

Specific points:

L39-44. No, the reason why we are uncertain about the effects of drainage and afforestation is because it is logistically hard to measure. Even whether drainage actually causes a loss of carbon is based more on expectation rather than hard measurement.

> We agree with this comment, and the preceding sentence (line XX-XX) states this quite clearly, We decided to remove the reference to numerical models, as our motivation is primarily to obtain data to establish drainage effects under forestry.

L97. The number of replicate plots is not given. Seems to be n = 4, repeated for each microform.

> Added in: "the four paired"

L135. Was the moss layer conidered as "litter" and removed also? The text suggests there is only litter, peat and tree roots, but the moss layer looks non-negligible in the photo.

> The moss layer was not removed. The site does have a sparse cover of mosses, we assume this only makes a small contribution to NPP, compared to the dense spruce/pine canopy and conifer needle input to surface litter.

L160. Soil moisture can be measured and expressed in many ways. Explain what is measured here - volumetric water content (m3/m3) by TDR method?

> Added in: "measuring m³/m³ (volumetric water content)"

L173. For clarity, it would be best to give the equation for the final fitted model. "plot as a random effect" could be either an intercept or a grouping term on one or more of the other coefficients. The former I think, from Table 2.

> Added: "(lmer(CO2 flux ~ (soil moisture *soil temperature * treatment * microform + treatment * microform * litter treatment) + (1|plot)))"

L179. The absolute values of AIC are very arbitrary, and there is no logic to saying that differences of less than 2 are meaningful. The relative values are meaningful, but there is no need to define such thresholds. The key thing is whether predictions differ substantially among these models - see point below.

The delta of less then 2 came from the referenced publication and we believe this is a useful way of comparing models. We appreciate that the referee has a different view on this.

L180+. Not sure why the weighting is mentioned, since it was not used. If there are notable differences between predictions from the different models, then using a weighted ensemble of model predictions would sensible.  Bayesian model averaging would be even better. If, however, predictions are all rather similar, that justifies the approach of choosing the single best model (minimum AIC).

It is mentioned to show how confident we are in the top model, smallest AIC and highest weighting.

L168/L185. Fitted with nlme, but predicted with lme4?  I think this is an error.

Indeed fitted and predicted with lme4, so changed in line 168.

L208. Only linear effects are considered here, but nonlinear effects are possible/expected, but harder to deal with and identify statistically.  Can we get some justification for this?

The $CO_2$ data is log transformed, so we do consider non-linear effects.

L210. 40 % of variation was explained by the model, but this is presumably on the log scale. It needs pointing out that predictions are made in the original units, and all the uncertainties reinflate.

Added: "on the log scale, since predictions are made in the original units all uncertainties reinflate "

L215. Be explicit about the interpreation & units here - I think these are intercepts and multipliers for CO2 flux on the log scale (log(umol m-2 s-1) / deg C)?

We don't think these need units, all numerical predictors were standardized to one standard deviation prior to analysis, so these fixed effects are used to interpret how big of an effect the particular predictor has with their standard error and p-value. We have added to the table heading the following to make this clearer: "All numerical predictors were standardized to one standard deviation prior to analysis."

L215. There is no term for "Microform = Furrow". Maybe this is the interpretation of the interept?

Yes correct, this is the interpretation of the intercept.

L228. The negative interaction term just means that the T coefficient decreases with SM.

Figure 4 visualises what the negative interaction term means, which we have written in the text. We believe the text is correct.

L232-233. This is confusing, as it sounds like a separate step has been done.  However, the whole rationale of fitting a model including soil moisture is precisely this - so that comparisons between treatments can be made, whilst accounting for differences in soil moisture.

This was indeed an extra step, and done for the model predictions only (so this is not about the model fit). By using the soil moisture data from just outside the plots when predicting the fluxes from the trenched plots, we took this artefact of the experiment away and were able to come up with a more accurate peat oxidation rate. This is explained in the Methods section: "The predictions were made over half-hourly measurements of soil moisture and soil temperature at 5 cm soil depth in all three microforms just outside the plots."  (line 194).

L235. Can you show the data as well as the fitted model?

This was plotted directly from the model output and we believe there is not an option to add in the data.

L238. For completeness, be explicit how heterotrophic fluxes are estimated - presumably as total - autotrophic.

Added: "The model prediction of heterotrophic respiration is calculated by subtracting the autotrophic flux from the total soil flux and includes emissions from decomposition of cut roots"

L243. Be explicit how the uncertainty on the annual sum is calculated.  The error terms in Eqn 2 all add, and have to transformed from log to original units.

All error terms where indeed back transformed to original units, as well as the actual outputs and error terms were propagated. This has been added in the methods (Line 193 and 200).

L293. The table caption is very confusing. It reads as if this is the decay of dead roots itself, not the flux from the plot after the dead-root correction. Needs re-wording.

Changed to: "Corrected for dead root decay in trenched plots, heterotrophic (peat only; Fh) and autotrophic (Fa) fluxes (standard error in brackets) in g C m-2 y-1"

L378+. Of course there have to be codicils on the results, and it is the C balance over the lifespan of the forest that matters. However, the expectation in most modelling work is that drainage and afforestation causes oxidation of the peat at a rate of 50 to 300 g C m-2 y-1 (e.g. Cannell 1993). This is offset in the first few rotations by the increasing tree biomass and litter, but ultimately, the ongoing long-term degradation of the peat becomes the dominant term, and the system becomes a net carbon source after 1-5 rotations (depending on the assumed peat oxidation rate). If this study is in fact showing that the peat is a net sink of 17 to 124 g C m-2 y-1 after 30 years, this is surely the stand-out result. Worth some more discussion at least.

We have expanded and in part re-written the discussion around this (also in response to another referee; see below).

---

## Author Comment (AC3)

**Referee 3**

This is a well-presented and relevant study about soil respiration partitioning and the soil carbon (C) balance of drained afforested peatlands in temperate climates. Although soil respiration studies are common nowadays, the amount of data from these ecosystems and climatic zone is still scarce. The long-term consequences of draining and afforesting peatlands with conifers is still under debate and contradictory results (i.e. soils being carbon sinks or sources) can be found between study sites and years. In addition, results from this soil C balance study could help developing stronger national Tier 2 emission factors for this land use in the UK and also increase the number of study sites and data used to develop Tier 1 emission factors from the 2013 Wetlands Supplement. However, I have two important concerns that would need to be addressed and further discussed in the manuscript. I think that, once these two potential issues have been revised, it would make a nice paper well worth publishing in Biogeosciences.

Thank you for your positive and supportive feedback.

**General comments**

Issue #1: soil C balance

The main issue I see in this manuscript is about the method used to calculate the soil C balance. In Line 61, it says that C inputs into the soil are represented by litterfall only. There is no mention to other C inputs such as organic matter from fine root and moss litter. If mosses are not present or they represent a very small fraction of the C inputs (lines 86 and 87), they should still be mentioned and a justification of why this has been omitted should be given. However, fine root litter (using the measured fine root biomass and an appropriate turnover rate) should be considered in the soil C balance because this is an important and significant C input. Not adding this C input would result in an underestimation of the soil C balance.

> This is a good point and we're happy to elaborate on it. The site does have a sparse cover of mosses, which we assume makes only a small contribution to NPP, compared to the dense spruce/pine canopy and conifer needle input to surface litter. We did not quantify root production and turnover, as this was beyond the scope of our study, although we acknowledge that 'litter input' includes belowground as well as aboveground litter. We have used needle litter fall as a directly measurable C source for soil organic matter input, but acknowledge that this is a very conservative estimate due to the omission of root (and to a small extent moss) inputs.

> We make this omission clear in the discussion (lines 330 – 336), where we also use literature estimates to provide outline estimates of the resulting under-estimation of C inputs, and highlight the fact that our estimated small carbon sink in these afforested peatland sols have to be considered conservative for that reason..

To facilitate the reader how this has been calculated, I would suggest adding an equation with all the components of the soil C balance and their uncertainty. Also, C outputs are represented by heterotrophic respiration and therefore, these fluxes should represent peat and litter decomposition. However, in multiple occasions, it is written as "heterotrophic (peat only) fluxes". When considering peat fluxes, please, mention it like that, peat respiration or peat fluxes and only use the "heterotrophic respiration" terms when both, litter and peat respiration are considered together.

This is a useful suggestion, and we have corrected our use of "heterotrophic fluxes" to refer to total fluxes (peat and litter) only, using "peat fluxes" or "peat decomposition" wherever appropriate.

In the discussion, line 309, in says that mass balance calculations indicate that soils are net sink of C but it does not specify any number (and ideally together with an error). In addition, this mass balance has not been presented in the methods neither in the results section. In my opinion, this mass balance calculation is one of the most important results from this study and therefore, it should be better explained and discussed.

Furthermore, from looking at figure 8, it seems that autotrophic respiration has been included in the soil C balance and this is not correct. This soil respiration component is not part of the soil C balance as this is not related to peat oxidation. This is part of the ecosystem respiration and net ecosystem exchange and it should be used to assess the net C balance of the plantation (i.e. when the C in the tree biomass is being considered) but it is not part of the soil C balance.

Figure 8 is a summary diagram using results presented in tables and text throughout the results section. As indicated above, we lack some terms of a complete C balance (root turnover), but even without this input, our results show a net sink of C in these soils. We now provide this estimate (with error) based on values in Figure 8. To avoid confusion, we have deleted the term "mass balance", and instead present the "soil surface C balance". This includes autotrophic respiration, as this is a key result of our study, aimed at partitioning soil $CO_2$ efflux. The following paragraph discusses values and clarifies the relevance of different terms to the soil C budget at our site.

Finally, the soil C balance is compared with results from Minkkinen et al (2018) which found that the drained peatland forest was a net soil C sink of -60 gC/m2/y (lines 344 to 347). However, this value from Minkkinen et al was derived using Eddy Covariance measurements. It would be more useful to compare the soil C balance with results calculated using similar methods like that from the same Minkkinen et al paper which is derived from chamber techniques. If using chamber techniques, Minkkinen et al reported that the site was a small soil C source. Similar results are also found in Ojanen et al.

Thank you for pointing this out. We agree that the Minkkinen et al (2018) paper shows a slight C source: Litter input = 437 g C $m^{-2}$ $a^{-1}$ (no error presented) vs. heterotrophic $CO_2$ efflux of 475 ± 31 g C $m^{-2}$ $a^{-1}$. We now indicate that the 'headline figure' of -60 g C $m^{-2}$ $a^{-1}$ is based on eddy covariance estimates, and that comparable chamber based estimate show a weak soil C sink.

Overall and as already pointed out, this is be the main objective of this manuscript and the method should be better described and the results and their implications further discussed. These results will define whether conifer plantations on drained peatlands are net soil C sources and sinks. Therefore, everything related to how this is calculated should be presented clearly. Some useful publications about soil C balance in forestry-drained and afforested peatlands:

Ojanen, P., Minkkinen, K., Lohila, A., Badorek, T. & Penttilä, T. 2012. Chamber measured soil respiration: A useful tool for estimating the carbon balance of peatland forest soils? Forest Ecology and Management, 277, 132-140.

Ojanen, P., Minkkinen, K. & Penttilä, T. 2013. The current greenhouse gas impact of forestry-drained boreal peatlands. Forest Ecology and Management, 289, 201-208 (**already cited**)

Ojanen, P., Lehtonen, A., Heikkinen, J., Penttilä, T. & Minkkinen, K. 2014. Soil CO2 balance and its uncertainty in forestry-drained peatlands in Finland. Forest Ecology and Management, 325, 60-73

Minkkinen, K., Ojanen, P., Penttilä, T., Aurela, M., Laurila, T., Tuovinen, J. P. & Lohila, A. 2018. Persistent carbon sink at a boreal drained bog forest. Biogeosciences, 15, 3603-3624 (**already cited**)

Jovani-Sancho, A.J., Cummins, T. and Byrne, K.A. (2021), Soil carbon balance of afforested peatlands in the maritime temperate climatic zone. Glob Change Biol, 27: 3681-3698

Thank you for listing fill references, this has been useful.

Issue #2: soil CO2 fluxes

My second concern is about the low soil CO2 fluxes reported in this study. As it can be seen in Figure 9, both, heterotrophic respiration and total soil respiration, are half or even up to three times smaller (for total soil respiration) than fluxes from boreal forest on peat soils. The reason behind why the measured fluxes in this study are that low should be further explored and discussed. While trenching produces many uncertainties on measured heterotrophic (as pointed out by the authors) total soil respiration should allow an easier comparison between fluxes from Sitka spruce plantations across different study sites and environmental conditions.

We have checked our flux calculations, and the rates are correct. We discuss this finding in context of literature reporting a range of fluxes and flux partitioning from temperate and boreal afforested peatlands. In this re-written section of the discussion, we address the referee's concern of methodological bias and include possible explanations or at least likely factors associated with an explanation for our relatively low flux sums. We note, however, that the flux ratios we report fit well with global patterns based on heterotrophic/total soil $CO_2$ flux, as shown in Figure 9.

In Lines 334 to 339, the authors compare the CO2 fluxes with results from Byrne and Farrell (2005) and Hargreaves (2003), studies with similar CO2 fluxes for total soil respiration and peat oxidation, respectively. Although Byrne and Farrell (2005) is a very nice and useful study, the method used to measure soil CO2 was based on soda-lime technique which is clearly not comparable with results from and infrared gas analyser like the EGM-4 used in the present study. These differences is the methods is highly relevant for potential readers and it should be clearly stated. In addition, there are other very interesting soil respiration partitioning studies (see Makiranta et al 2008) or heterotrophic respiration from drained peatland forests (Minkkinen et al 2007) that could be used to compare the peat, litter and root respiration values. While comparisons with results from Jovani-Sancho et al (2018) only focused on total soil respiration, other useful results from peat and litter respiration are provided in such study. Yamulki et al also provides useful soil respiration data for drained afforested peatlands with lodgepole pine. I would suggest a broader comparison with other

soil respiration studies on both, temperate and boreal peatlands. It is likely that such comparisons with the mentioned studies (or others selected by the authors) would show large differences in peat respiration.

> The referee raises good points here. We now place our flux estimates incontext of literature. We decided to drop the reference to Hargreaves et al. (2003) (the only study reporting lower fluxes than ours); the authors used modelling to estimate flux contributions, but we were unsure about their site comparison to derive autotrophic and heterotrophic contributions.

My question is , could all these differences be explained by the artefact of the dead root biomass and not having applied the "C flux from dead roots" correction? This is briefly mentioned in line 350-351. Could this flux correction be applied to one or two studies and see how the soil CO2 fluxes would vary?

> We have included the issue of root decomposition and its effect on our estimates. Rather than attempting to estimate root decomposition derived fluxes for other studies, we highlight the magnitude of flux reduction that we applied to provide this information.

Finally, something to point out is that the reported total soil respiration (342.5 gC/m2/year; lines 334 and 336) are much lower than modelled total soil respiration (between 556 and 991 gC/m2/year) a Sitka spruce chronosequence on mineral soils in Ireland (see Saiz et al 2006). This could perhaps be explained by the fine root biomass, climate or nutrient content. But also, modelled heterotrophic respiration from the same Saiz et al study (between 240 and 403 gC/m2/year) seems to be much higher than heterotrophic respiration from Hermans et al (115 gC/m2/year). I would imagine that heterotrophic respiration would be greater in a drained peatlands than in wet mineral gley soils.

> We appreciate the referee's comment here. After consideration, we decided to not broaden the comparison of flux partitioning to mineral soils, as this would open up a lengthy discussion of other literature from mineral soil based forestry, which is outside the scope of our study.

I would suggest a final check on the flux calculations to make sure that everything is correct. In line 121 says that collars of 10 cm collars were inserted 3 cm into the peat. Were the remaining 7 cm of the collar added to the 5 cm of the chamber when calculating the chamber's headspace? If so, the total dimensions would be a height of 12 and a diameter of 20 cm. Does this diameter refer to the internal dimensions of the chamber? Knowing the exact dimensions and volume of the chamber would be useful. And finally, could the 3 cm insertion depth have sever some of the fine root located at the top of the floor surface? Did you have surface roots (below or growing through the fresh litter) on your study sites Sitka spruce on afforested peatland have most of the fine roots located on the top cm of the soil. Please, see Heinemeyer et al 20011 and Jovani-Sancho et al 2017 for peatland-specific studies about this effect and Jian et al 2020 for a global review.

> We have checked our flux calculations again, including chamber dimensions, and are confident that there is no error. We are familiar with the issue of root cutting and hence potential artefacts of reduced $CO_2$ flux. The insertion depth was superficial (3 cm is a maximum estimate), and would have been no barrier to root re-growth under the collar in case of damage when collars were

> established. Note also that collar insertion does not explain the low decomposition estimates, as there should be no impact in girdled plots (as roots are cut anyway).

Mäkiranta, P., Minkkinen, K., Hytönen, J. & Laine, J. 2008. Factors causing temporal and spatial variation in heterotrophic and rhizospheric components of soil respiration in afforested organic soil croplands in Finland. Soil Biology and Biochemistry, 40, 1592-1600.

Minkkinen, K., Lame, J., Shurpali, N. J., Mäkiranta, P., Alm, J. & Penttilä, T. 2007. Heterotrophic soil respiration in forestry-drained peatlands. Boreal Environment Research, 12.

Saiz, G., Byrne, K. A., Butterbach-Bahl, K., Kiese, R., Blujdea, V. & Farrell, E. P. 2006. Stand age-related effects on soil respiration in a first rotation Sitka spruce chronosequence in central Ireland. Global Change Biology, 12, 1007-1020.

Collar insertion effects

Heinemeyer, A., Di Bene, C., Lloyd, A.R., Tortorella, D., Baxter, R., Huntley, B., Gelsomino, A. and Ineson, P. (2011), Soil respiration: implications of the plant-soil continuum and respiration chamber collar-insertion depth on measurement and modelling of soil CO2 efflux rates in three ecosystems. European Journal of Soil Science, 62: 82-94.

Jovani-Sancho AJ, Cummins T, Byrne KA (2017b) Collar insertion depth effects on soil respiration in afforested peatlands. Biol Fertil Soils 53:677–689.

Jian, J., Gough, C., Sihi, D., Hopple, A. M., & Bond-Lamberty, B. (2020). Collar properties and measurement time confer minimal bias overall on annual soil respiration estimates in a global database. Journal of Geophysical Research: Biogeosciences, 125, e2020JG006066.

> These references have been very useful, and we appreciate the supportive approach of this referee to improve the paper.

---

## Referee Report (RR1)

**Review of MS BG-2021-126**

By Russell Anderson

**Specific comments**

1. The amended title is now ambiguous. It could be read as being about separating soil $CO_2$ effluxes (both autotrophic and heterotrophic) from net soil carbon balance. Even though that would not make much sense, you should remove the ambiguity, perhaps by rewording it as 'Net soil carbon balance in afforested peatlands and separating autotrophic and heterotrophic soil $CO_2$ effluxes'.
2. Lines 438-439. You state 'In the UK forest plantations on deep peat usually end in clear felling of the site and restoration of the peat.' In fact it is still more common for UK forests on deep peat to be restocked for a further forestry rotation than for the sites to be restored to open peatland. This could be corrected simply by changing 'usually' to 'sometimes'.

---

## Author Response (AR2)

1. Line 19: "large change in the heterotrophic : autotrophic fraction" might be clearer

Changed "big" to "large".

2. L. 88-89: italicize species names

Italicized the following names in line 88-89: "*Hypnum jutlandicum, Hylocomium splendens Sphagnum fallax* and *S. capillifolium*"

3. L. 167: Specify what version of R, and update citation accordingly

The version is in the reference list, in the full citation of Rstudio (1.0.136). After googling it seems that that is the accepted way of referencing the version, but we're happy to add it in the "in text reference" if that would be better.

4. Figure 3: axis label says "month and year" but the tick labels are DD-MM-YY, which is slightly confusing

Changed the x-axis label to "Date".

5. Figure 5: what is the fan of straight lines in each panel?

The fan of straight lines is due to an artefact and have been removed from the graphs.

6. Data availability (line 415): this is not adequate, as the journal policy is " If the data are not publicly accessible, a detailed explanation of why this is the case is required." Please elaborate.

The data is currently being made publicly available through STORRE (https://storre.stir.ac.uk/). We can send the link over as soon as it is available.

**Referee 1**

1. The amended title is now ambiguous. It could be read as being about separating soil CO2 effluxes (both autotrophic and heterotrophic) from net soil carbon balance. Even though that would not make much sense, you should remove the ambiguity, perhaps by rewording it as 'Net soil carbon balance in afforested peatlands and separating autotrophic and heterotrophic soil CO2 effluxes'.

We have changed the title to: "'Net soil carbon balance in afforested peatlands and separating autotrophic and heterotrophic soil $CO_2$ effluxes"

2. Lines 438-439. You state 'In the UK forest plantations on deep peat usually end in clear felling of the site and restoration of the peat.' In fact it is still more common for UK forests on deep peat to be restocked for a further forestry rotation than for the sites to be restored to open peatland. This could be corrected simply by changing 'usually' to 'sometimes'.

We have changed "Usually" to "sometimes".